# The Impact of Health and Social Care Professionals’ Education on the Quality of Serious Illness Conversations in Nursing Homes: Study Protocol of a Quality Improvement Project

**DOI:** 10.3390/ijerph20010725

**Published:** 2022-12-30

**Authors:** Silvia Gonella, Paola Di Giulio, Paola Berchialla, Mario Bo, Paolo Cotogni, Giorgia Macchi, Sara Campagna, Valerio Dimonte

**Affiliations:** 1Direction of Health Professions, City of Health and Science University Hospital of Torino, Corso Bramante 88-90, 10126 Turin, Italy; 2Department of Public Health and Pediatrics, University of Torino, via Santena 5 bis, 10126 Turin, Italy; 3Department of Clinical and Biological Sciences, University of Torino, via Santena 5 bis, 10126 Torino, Italy; 4Section of Geriatrics, Department of Medical Sciences, University of Torino, City of Health and Science University Hospital of Turin, Corso Bramante 88-90, 10126 Turin, Italy; 5Pain Management and Palliative Care, Department of Anesthesia, Intensive Care and Emergency, University of Torino, City of Health and Science University Hospital of Turin, Corso Bramante 88-90, 10126 Turin, Italy

**Keywords:** aged, blended learning, communication skills, education, multimethod studies, nursing home, patient-centered care, serious illness conversations, quality of care, training

## Abstract

Health and social care professionals (HCPs) who work in nursing homes (NHs) are increasingly required to sustain serious illness conversations about care goals and preferences. Although these conversations may also be challenging for experienced HCPs and the literature recognizes high-quality communication as key to providing patient-centered care, so far, no specific educational program has been developed for the NH setting to improve HCPs’ communication skills. Our study aims to test the feasibility and potential effectiveness of an innovative, blended communication skills training program (*Teach-to-Communicate*) targeting the HCPs who work in NHs. This program includes classroom-based theory, experiential learning, and e-learning, and relies on interdisciplinary contexts and several didactic methods. The study consists of two phases: phase I is the development of written resources that employ focus group discussion involving field experts and external feedback from key stakeholders. Phase II consists of a multicenter, pilot, pre-post study with nested qualitative study. The *Teach-to-Communicate* training program is expected to enhance the quality of communication in NH and HCPs’ confidence in sustaining serious illness conversations, reduce family carers’ psycho-emotional burden and improve their satisfaction with the care received, and increase advance care planning documentation. Our protocol will provide insight for future researchers, healthcare providers, and policymakers and pave the way for blended educational approaches in the field of communication skills training.

## 1. Introduction

Conveying difficult news is common in clinical practice across several settings, including oncology, intensive care, palliative care, or nursing homes (NHs), and can become even more challenging at the end of life [1].

End-of-life communication in NHs has been receiving increasing attention in recent years due to its potential to influence the quality and intensity of care at the end of life [2]. Poor or absent communication increases the risk for NH residents to receive unwanted, aggressive, and not beneficial medical care during the last period of life, which is associated with residents’ deterioration in the quality of life and death and family carers’ psycho-emotional burden and dissatisfaction with the care received [3]. Instead, good communication promotes shared decision-making, aligns care with a person’s preferences and values, and reduces aggressive care at the end of life [2,4]. Timely discussion of end-of-life issues improves residents’ and their family carers’ acceptance of the illness and sustains the transition from curative-oriented to palliative-oriented care with the primary goal to relieve pain and discomfort rather than prolonging life [5]. 

Unfortunately, despite the unquestionable benefit of high-quality end-of-life communication, health and social care professionals (hereafter, healthcare professionals, HCPs) often fail to engage in serious illness conversations. HCPs may feel underprepared to answer some challenging family carers’ questions, such as how much time is left for their relative, due to prognostic difficulties and limited training [6,7,8]. Moreover, these conversations represent a source of distress for HCPs who may experience a wide range of unpleasant emotions, including anxiety and frustration [9]. Thus, the tendency is to avoid or postpone the conversations until the recovery is impossible, with most conversations taking place in the last month before death [10]. Beyond being late, serious illness conversations are brief, infrequent, and limited to addressing prognosis and disease trajectory, while psychological, spiritual, and existential problems are much less frequently discussed [2,11].

Several educational interventions have been developed to train HCPs in end-of-life communication over the past three decades [12,13], but none specifically targeted HCPs working in the NH setting. Moreover, most of these programs were monodisciplinary and developed for physicians or residents [13], while only a few enrolled other healthcare or psychosocial professionals [14,15,16], even if educational needs for communication skills are common among HCPs irrespective of scope of practice [7,17,18] and interdisciplinary training programs appear promising [19]. An interdisciplinary learning environment in which HCPs can benefit from group feedback and reflect on implications of difficult conversations entail, as well as habits in communication, improved HCPs’ confidence in engaging in serious illness conversations, communication skills, self-reflective attitudes, and commitment to interdisciplinary collaboration to manage difficult conversations [14,20]. Additionally, discussing with colleagues from different disciplines allowed them to improve relational skills, such as empathy and creativity, that are as important as knowledge of communication theories and protocols to successfully conduct difficult conversations [19,21,22].

Previous training programs relied mainly on traditional didactic methods—lectures, small group discussion, brainstorming sessions on successful and failed communication, reflection, problem-based learning techniques, role play, and hands-on training [12,23]—and explored HCPs’ self-reported outcomes [12]. Self-efficacy, knowledge, and communication skills were the most common measures with evidence of improvement; however, of very low quality due to the overall high risk of bias [13]. Furthermore, experts in the field of communication skills argued for different training delivery methods [24] and the need for evidence on the effectiveness of interventions on patient outcomes [13].

Additionally, the COVID-19 pandemic highlighted the need for specialized training in technology-based communication in addition to in-person interactions [25]. Establishing trusting relationships and building meaningful dialogues in virtual settings may be complex and impeded by several obstacles, including little or limited bodily communication and an increased risk of misunderstanding [26]. Therefore, it is crucial to develop educational initiatives that aim to increase the communication skills of HCPs and their confidence to handle conversations about serious diseases both in person and in remote settings. 

This project aims at setting up and evaluating the feasibility and effectiveness of a blended educational communication skills training, targeted at the HCPs who work in the NH setting, that includes traditional didactic methods and e-learning. The intervention is framed within an innovative pedagogical approach that employs experiential methodologies and integrates the teaching of specific theoretical communication models with the promotion of self-reflection and relational abilities in an interdisciplinary context [27]. Our study hypotheses are that the training program will produce: (a) better quality of communication reported by family carers and HCPs; (b) greater HCPs’ confidence to sustain both in-presence and technology-based serious illness conversations; (c) improved family carers’ satisfaction with the care received; (d) reduced family carers’ psycho-emotional burden; and (e) increased completion of any advance care planning (ACP) documents.

## 2. Research Methods and Analysis

### 2.1. Study Design

The study protocol was designed following the standard protocol items [28]. The pilot feasibility study will follow the Transparent Reporting of Evaluations with Nonrandomized Designs (TREND) guidelines [29]. Qualitative data will be reported following the Consolidated Criteria for Reporting Qualitative (COREQ) research checklist [30]. 

Figure 1 outlines the two project phases and related inscribed actions: (phase 1) development of written resources; and (phase 2) multicenter, pilot, pre-post study with nested qualitative study. 

#### 2.1.1. Phase 1. Development of Written Resources

The first project phase involves the development and assessment of written resources (i.e., a booklet) aimed at providing guidance to HCPs in initiating and sustaining serious illness conversations (Figure 1):

Box 1Objectives of the Pilot Pre-Post Study➢To determine how many healthcare professionals attend the training program compared to the eligible.➢To determine how many participating healthcare professionals complete the 6-month training program.➢To explore when the family meetings are conducted (i.e., working time vs off duty) and estimate the number of family meetings over a 6-month period.➢To determine how many family carers of nursing home residents participate in the study.➢To estimate refusal rate and 6-month follow-up rates.➢To assess, qualitatively, the intervention delivery, implementation barriers, and suggestions for improvement.➢To estimate the quality of communication perceived by family carers at 6-month follow-up (primary study outcome).➢To assess changes in secondary outcome measures.➢To inform the sample size estimation for subsequent studies.

##### Development of the Booklet

A co-creation process will be employed to develop a booklet. A first draft of the booklet was arranged, which presented the most frequent and challenging communication scenarios according to the literature [6,8] and previous interviews with HCPs having different scopes of practice [7]. The booklet is 99 pages in length and about half of it consists of worksheets. Each scenario is evidence-based and accompanied by evidence-based indications to establish quality communication and overcome communication challenges. Secondly, three focus group discussions will be held involving a bioethicist, a psychologist, two social workers, an internist, an emergency physician, two palliative care physicians, a home palliative care nurse, two medicine nurses, an NH nurse, an NH manager, a forensic scientist, two representatives of associations of patients and family carers with life-limiting illnesses, and a representative of a local volunteer association supporting family carers of persons with dementia. Each focus group will involve 5–6 participants and take place remotely under the moderation of a member of the research team. Discussion will be encouraged using a pre-defined list of questions. Another team member will pick up in-the-field notes concerning participants’ demeanor and para-verbal language. Discussions will be recorded after consensus is reached by each participant. Each participant will receive the booklet 7–10 days before the focus group to ensure that they have enough time to read it. Thirdly, the booklet will be refined according to the suggestions that emerged from the focus groups and sent out for a second evaluation with the request to fill in a 10-min questionnaire aimed to assess the booklet’s content (i.e., comprehensibility, completeness, missed information, need for improvement) and editing (i.e., length, visual presentation, readability). Fourthly, the booklet will be further adjusted according to the emerging comments and sent out for external review by health and social care professionals not involved in its development. Moreover, to ensure genuine participation, the views and preferences of at least two broad societies will be sought. Stakeholders will be contacted by email and letters as appropriate, and their feedback will be sought via a brief open-ended questionnaire. Finally, feedback from external consultation will be assimilated and the final version of the booklet published. 

##### Assessment of the Booklet: Acceptability, Usefulness, and Feasibility to Use in Daily Practice

Acceptability, perceived usefulness, and feasibility of using the booklet in daily practice as perceived by HCPs will be assessed. Acceptability and feasibility will be evaluated at 4 weeks after the implementation of the booklet, and usefulness at 4 weeks and 6 months. It will be possible to suggest changes. 

Acceptability is referred to by O’Connor and Cranney as ‘‘ratings regarding the comprehensibility of components of a decision aid, its length, amount of information, balance in presentation of information about options, and overall suitability for decision making” [31]. Our measure of ‘‘acceptability’’ refers more specifically to acceptability to use, understand, read, and inform families. It also includes the themes of communication with families, relationships with families, and the intention to use them in practice. For this, we developed a 10-item scale summing the rating of agreement with statements selected from the acceptability instrument by O’Connor and Cranney [31]. Item scores were 1 (strongly disagree) to 5 (strongly agree) [31]. Higher total scores represent better acceptability. Validity, reliability, and internal consistency (Cronbach’s α) of the scale will be verified. 

Usefulness and feasibility of use will be rated on a scale from 1 (not at all useful/feasible to use) to 10 (extremely useful/feasible to use). 

#### 2.1.2. Phase 2. Multicenter Pilot Feasibility Pre-Post Study with Nested Qualitative Study

The second project phase will be dedicated to the conduct of the multicenter pilot feasibility pre-post study with nested qualitative study in four NHs. This phase has three inscribed actions—intervention set up, pilot study, and qualitative study (Figure 1)—and aims to test the feasibility and potential effectiveness of the educational intervention.

**(i)** 
**Intervention set up**


##### Training Program

The goal of the training program is to equip HCPs with communication abilities to sustain serious illness conversations. To achieve this, HCPs will attend a training program (called *Teach-to-Communicate*) that will be Continuing Medical Education accredited. The program will develop over 6 months with a duration of 20 h. The program aims to improve HCPs’ skills and competencies in serious illness conversations based on updated scientific evidence; guide HCPs in overcoming challenging communication scenarios, including technology-based communication; and improve communication between HCPs and residents/families in clinical practice to promote an effective partnership. 

The training will employ different methodologies and consist of three learning modules:First module (6 h, residential event): (a) a 2-h lecture on the clinical and ethical principles of end-of-life communication, general and situation-specific communication strategies, and content and use of the communication booklet (output of phase 1); (b) a brainstorming session on successful and failed communication; (c) videos pointing out different communication challenges and communication skills abilities; (d) a small group-based reflective discussion.

The videos will be based on realistic, complex clinical scenarios that are relevant to the learners’ clinical setting and their role. Staff are unlikely to make efforts to apply learning in practice if training is perceived to lack relevance and applicability [32]. Two scripts will be developed for each video: one script in “good form”, where HCPs employ appropriate communication strategies to overcome challenges, and one script in “bad form”, showing interactions of poor quality. Bad forms will be shown first, and participants will be asked to share feedback and explain what they would do differently. A mixed version (good and bad communication examples in the same video) was excluded to promote discussion and learning [33]. Professional actors will be involved. 

2.Second module (within 3 months from the first module, 4 h):
a.Experiential workshop based on improvisational theater techniques (3 h) [34]. Trained team members acting as HCPs and patients will role-play difficult conversations. The scripts will be based on previous interviews with HCPs who work in NH to identify real-world situations. A facilitator will introduce the actors and encourage participants to note parts of the dialogue that do not help us overcome difficult conversations. Participants will be asked for their feedback, including specific parts of the dialogue that could be improved. The scene will be reenacted, and participants will be encouraged to shout “stop” if they want to change the HCP dialogue. When a participant yells “stop”, the performance will be suspended, and the facilitator will ask the person the reason(s) why they have interrupted the performance and to assume the role of the HCP and use the suggested approach, becoming a spectator-actor. After the performance, the facilitator will ask the spectator-actors if they have carried out what they planned and ask the audience to comment on the performance of the spectator-actors. The critics of the performance will be asked to explain what they would do differently and will be invited to play the encounter, including their suggestions. These scenarios will be repeated several times to allow most participants to practice these difficult conversations.b.Community of practice I (1 h): at the end of the experiential workshop, a member of the care team will facilitate a community of practice. Participants will share the feelings and challenges they experienced while playing the encounter. The facilitator will foster reflection on the difficulties in applying suggestions and communication protocols effectively in a real situation.3.Third module (from 3 to 6 months from the first module, 10 h):
a.On-site training (2 h): participants will conduct at least two family meetings in their clinical setting in a quiet room using the communication booklet. The care team will identify those family carers who could benefit from a family meeting and its goals (for example, communication of their relative’s worsening conditions). Family meetings must be planned and agreed with family carers in advance and may be conducted during the working time because they are part of good clinical practice; no minimum or maximum duration is set, and the duration depends on the specific situation.b.Community of practice II (4 h, in-presence or technology-based according to the facility preferences): a member of the care team will facilitate the community of practice. Participants will share problems that emerged during the family meetings and the strategies employed to overcome them in a community of practice encounter, followed by collective reflection. This final session is reserved for those who have completed all activities including at least two family meetings and self-learning activities (see paragraph web platform). Each participant will self-certify that they have conducted the minimum number of family meetings required.

Trainers will be a panel of nurses, a psychologist, a palliative care physician, and a bioethicist. All have experience leading training courses and workshops on HCP-patient/family communication. 

##### Web Platform 

As part of the intervention set up, a web-based platform will be created for complementary, asynchronous, online, self-education activities that will be an integral part of the intervention (4 h). A web-based platform will be created and made accessible for the overall 6-month training program. Asynchronous training will consist of videos showing family meetings and communication skills in both good and bad versions. Each video will be supported by a critical comment explaining communication mistakes (bad form) or the communication strategies used (good form). Additional reading material will also be available, as well as a forum where participants can interact and exchange opinions and reflections. A psychologist will be available for the overall 6-month training program to moderate online discussions and provide in-person support, if needed. 

##### Linguistic Validation of Measures

Two outcome measures not available in Italian will be translated and cross-culturally adapted [35]: the family carer version of the 13-item Quality of Communication Questionnaire (QOC) [36] and the satisfaction with care at the end of life in dementia (SWC-EOLD) questionnaire [37]. The main steps will be as follows [35]:

Forward translation: two qualified Italian translators who are culturally representative of the target population will produce two independent translations. The two translators will have different backgrounds: one will be aware of the concepts being examined, while the other will be a naive translator with no clinical background.

Synthesis of the translations: The translators and a team member will review the forward translations against the original version and produce a consensus version.

Backward translation: the consensus translation will be independently translated back into the source language by two mother tongue translators blind to the original version. The two translators will not be informed of the concept explored. 

Expert committee review: A panel consisting of the translators (forward and back translators), a methodologist, two HCPs, and two lay persons will review all the translations and the original questionnaire and reach a consensus on any discrepancies. A pre-final version will be produced.

Test of the pre-final version: each translated questionnaire will be proofread and then debriefed with at least 20 family carers and a panel of experts. 

**(ii)** 
**Pilot pre-post study**


Four NHs will be purposefully involved in the pilot feasibility pre-post study and sampled for geographical area and size to ensure the greatest variation of data. All NHs will be located in the northwest of Italy and will participate on a voluntary basis. Participation will be promoted through a local nursing home association network. 

There will be a baseline assessment (T0), the beginning of the training program within 1 month of the baseline assessment, and a follow-up assessment within 4 weeks of the initiation of the training program (T1) and 6 months (T2) thereafter. The baseline and follow-up assessments will be performed via anonymized records filled in by paper and pencil. A member of the research team will input the collected data into a password-protected database accessible only to the team members.

Participants (NH staff, family carers) will be free to withdraw from the study at any time, without giving reasons and without consequence. Information about drop-out reasons will be collected. 

The objectives of the pilot feasibility pre-post study are reported in Box 1. The study procedures are shown in Figure 2.

##### Eligibility and Screening

HCPs (a) of any profile, (b) with at least 6 months experience in the facility, (c) responsible for serious illness conversations with family carers, and (d) willing to participate in the study, will be eligible. Administrative staff will be excluded as their role does not involve clinical conversations with family carers. 

The main family contact for all institutionalized residents will be included if they are: ≥18 years, capable of communicating in Italian, and gave their written consent, regardless of the point of care of their relative (i.e., transition phase to the nursing home [38], deterioration-in-condition phase [39], or end-of-life phase with death expected within the next few weeks or within a few months [40]).

The number and profile of HCPs to be trained will be agreed upon with each participating facility according to their internal processes of communication with family carers. Anyway, it will be recommended to train at least three, four, and five HCPs in facilities with less than 50 beds, 50 to 100 beds, and more than 100 beds, respectively. No minimum nor maximum number of family carers per NH will be prespecified.

##### Baseline Assessment (T0)

A reference person will be identified in each NH. This person will identify eligible HCPs and family carers, provide written information about the purpose and procedures of the study, and obtain signed informed consent forms, according to the Declaration of Helsinki and the Good Clinical Practice guidelines of the EU. The informed consent is kept on file by the study personnel and is available for inspection purposes.

Family carers will be approached when they come to visit their relative, while HCPs will be approached before starting or at the end of their working shift. Family carers and HCPs who adhere to the project will fill in a baseline questionnaire on paper, which will be returned to the reference person. The completion time is about 10 min for the HCP questionnaire and 15 min for the family carer questionnaire. The reference person will collect data from clinical records about any hospital services used by residents in the last month (i.e., type of service, reason(s) for using, and number of day(s)/attendances) and if any ACP documents were in place (i.e., advance care plan, advance decision to refuse treatment, do not hospitalize, do not resuscitate, do not intubate, no artificial feeding, no artificial hydration, no invasive diagnostic procedures, comfort-oriented care, power of attorney). 

A set of variables will also be collected to describe the context of the facility (Table 1). NH managers will be asked to fill in a semi-structured ad hoc questionnaire exploring structure variables (e.g., public or private NH and staffing) and process variables (e.g., written procedures, communication resources, and meetings with family carers). HCPs will be asked to report on their relationships with colleagues and the leadership. The relationship with colleagues will be assessed using the interpersonal [41] and the group cohesiveness questionnaire [42]. The former shows a set of seven Venn-like diagrams, each representing different degrees of overlap of two circles, and HCPs have to select the picture that best describes their relationship with colleagues [41]. The latter is a questionnaire of eight items, each rated on a 7-point Likert scale from 1 (“strongly disagree”) to 7 (“strongly agree”) [42]. The relationship with the leadership will be assessed using: (1) the leader identity granted by followers to the leader, an instrument of 10 items each rated on a 7-point Likert scale from 1 (“strongly disagree”) to 7 (“strongly agree”) [Lord and Gatti, Personal communication, 2018]; (2) Liking—The Leader Affect Questionnaires, a 5-item questionnaire rating each item on a 7-point Likert scale from 1 (“strongly disagree”) to 7 (“strongly agree”) [43]; and (3) followers’ satisfaction in their role that consists of seven items rated on a 6-point Likert scale from 1 (“strongly disagree”) to 6 (“strongly agree”) [44]. 

Each NH will collect information on the number of HCPs willing to participate in the educational intervention, the number of family carers eligible, approached, and involved in the project with reasons for refusal, the number of family meetings performed over the 6-month period, and when they were performed (i.e., working hours or off duty). 

Table 1 shows the variables aimed at describing the context of the facility with their assessment tools.

##### Educational Intervention 

The educational intervention is scheduled at the university headquarters (residential event(s), experiential workshop), the NH site (on-site training), and online (asynchronous self-learning activities and potentially a community of practice) and will start within a month since the baseline assessment. It will unfold during the study. For details, see the paragraph Intervention set up. 

##### Follow-Up Assessment (T1, T2)

HCPs and family carers will complete the paper questionnaires at 4 weeks (T1) and 6 months (T2) since the first learning module (assessment time of about 10 and 15 min for HCPs and family carers, respectively). The reference person will collect data from clinical records about any hospital services used by residents with associated reasons and any ACP documents in place (assessment time of approximately 10 min). 

##### Outcome Measures 

A range of measures will be collected to capture the effect of the *Teach-to-communicate* training program at multiple levels:

Family carers: quality of communication, burden, quality of care, satisfaction with the care received;

HCPs: quality of communication, confidence in and obstacles to serious illness conversations;

Residents: completion of any ACP documents.

The primary outcome is the quality of communication perceived by family carers at T2. Outcomes with their assessment tools are reported in Table 2. 

##### Quality of communication

We will assess the quality of communication considering two perspectives: HCPs and family carers. At T0, T1, and T2, family carers will complete the family carer version of the QOC questionnaire [36] and HCPs the HCP version. The family carer/patient version of the QOC was developed using quantitative and qualitative methods and is available in two versions (17-item and 13-item). In this study, we will use an adaptation of the 13-item version measuring general communication (six items) and communication about end-of-life care (seven items), each rated on a scale from 0 (“very worst I can imagine”) to 10 (“very best I can imagine”) or identified as something the HCP did not do (“did not do”) or that family carers were unsure of how to rate the HCP on a particular skill (“don’t know”). The 0/10 ratings are recoded to 1/11, with 0 imputed for “did not do”. 

In addition to the QOC, family carers will report self-perceived communication quality on a scale from 0 (“very worst”) to 10 (“very bad”). 

The HCP version of the QOC consists of two items exploring the confidence of HCPs in talking about dying and the quality of communication they perceive to provide, each rated on a scale from 0 (“at all/very worst”) to 10 (“extremely comfortable/very best”). “Don’t know” option is available. 

##### Other Outcome Measures 

Family satisfaction with care at the end of life will be assessed using the SWC-EOLD. This consists of 10 items (α = 0.90) and each scored on a 4-point Likert scale [37]. Moreover, we will assess overall satisfaction with care, satisfaction with medical care, and nursing care on a 5-point Likert scale ranging from “at all” to “very much”.

Family carers burden will be assessed using the ZBI [46], a 22-item self-report measure of subjective burden among family carers that addresses functional and behavioral impairments as well as the home care circumstances. In this study, we will remove items 11 and 13 since they specifically refer to the care home environment and, therefore, they are not applicable to the NH setting. Each item was scored on a 5-point Likert scale ranging from 0 (never) to 4 (nearly always present). Therefore, a 20-item tool with a total score of 0 (low burden) to 80 (high burden) will be used. 

Family carers’ self-reported care quality and HCPs’ confidence in communication will be assessed on a 5-point Likert scale ranging from “at all” to “very much”.

Semi-structured ad hoc questionnaires will be employed to assess hospital services used by residents, with their associated reasons, and any ACP documents.

Semi-structured interviews with both HCPs and family carers will inform on the intervention quality, its implementation barriers, and suggestions for improvement (for details, see the paragraph nested qualitative study).

##### Kick off and over-the-Project Meetings

There will be three study meetings based on remote modalities (video calls): the kick-off meeting will be held before baseline data collection (T0) starts. Participants will be two members of the research group, the NH managers, and the reference persons responsible for data collection in each facility. The purpose of this meeting is to provide clear information on the objective and procedure and to guide participants through the study documentation. The second meeting will be held the week before T1 data collection to discuss possible doubts, top-up the motivation of NHs, and provide a safe place for peer discussion on the implementation of the intervention. The third meeting will take place the week before T2 data collection to refresh the study aim, discuss possible difficulties that arise, and reflect on potential solutions. All meetings will last about 1.5–2 h. Additional meetings will be organized according to the needs of each NH. In addition, a reference person from the research team will be available for inquiries. 

**(iii)** 
**Nested qualitative study (T3)**


Eight weeks after the T2 follow-up assessment, family carers and HCPs involved in the *Teach-to-Communicate* training program will be involved in one-on-one, semi-structured, face-to-face or remote-based interviews according to their preferences (T3). The T2 questionnaire for both family carers and HCPs will ask for their willingness to be interviewed along with a reference contact (email or telephone). Interviewees will be identified using a maximum variation strategy. A minimum of 10 interviews with family carers and 10 interviews with HCPs will be held; the final number will depend on the achievement of “data saturation” [47]. 

The interviews will work as a process measure by informing participant perceptions of the quality of the training program, its impact, and its refinement. They will last no more than one hour and be led by members of the research team trained in qualitative research and not involved in the *Teach-to-Communicate* training program to reduce social desirability response bias. Open-ended questions with probes (e.g., what, which, why, how) will be employed to achieve an in-depth understanding of the experience and body language noted. Participants will be reassured that interviews are confidential and will be anonymized; moreover, information about the purpose of the study and the opportunity to stop the interview at any time and for any reason will be provided and written consent will be obtained. Before ending the interview, participants will have the opportunity to add any further thoughts or reflections. 

The interview with family carers aims to explore the quality of communication with HCPs in the last two months, perceived changes in the quality of communication, and/or the quality of care provided to the relative after the NH adhered to the project. The interview with HCPs will involve the HCPs engaged in the *Teach-to-Communicate* training program and is aimed at collecting lived experiences about the intervention and its impact on the quality of communication with family carers, possible barriers and facilitators to implementation, and suggestions for changes to ameliorate the intervention. 

### 2.2. Study Power

A formal sample size calculation is not required since this is a pilot feasibility study. However, by hypothesizing a 2-point change and moderate effect size (γ = 0.60) on the QOC questionnaire filled in by family carers at T2, with a power of 0.85, 2-sided α = 0.05, and 20% of family carers lost at T2, we aim to recruit at least 40 family carers. The 2-point change on the QOC questionnaire was based on the hypothesis that the intervention would improve at least two QOC items by 1 point [16]. 

### 2.3. Data Analysis

#### 2.3.1. Quantitative Data

Descriptive statistics will be employed. Categorical variables will be summarized as numbers and percentages, while continuous variables will be summarized as mean and standard deviation (SD), or median and interquartile range (IQR). Categorical variables will be compared using the χ^2^. The Shapiro–Wilk test will be adopted to test the normality assumption of continuous variables. Depending on the data distribution, the paired t-test or the Wilcoxon signed-rank test will be carried out; correlations will be computed using Pearson’s or Spearman’s coefficients. As an additional analysis of the temporal change in the primary outcome (i.e., the quality of communication as perceived by family carers), a linear regression analysis using time as a covariate will be used to test for a trend in the quality of communication. As the analysis will be performed at the family carer level, the Huber–White estimator will be used to adjust for the correlation between the multiple observations contributed by the same NH. A significant coefficient of the time variable will indicate that the primary outcome changes with respect to time. The goodness of fit will be evaluated by R^2^.

Change in the secondary outcome measures will be calculated, as well as the following feasibility outcomes: recruitment rate of HCPs engaged in the training program (and reasons for non-enrolment); retention rate (proportion of HCPs completing the training program; proportions of family carers and HCPs completing follow-ups); missing data (proportion fully completed for each scale at each time point). In the case of missing values, multiple imputation will be performed. A p value less than 0.05 will be considered statistically significant. All analyses will be performed using SPSS v. 27. 

#### 2.3.2. Qualitative Data

Interviews will be audio-recorded and transcribed verbatim. Data analysis will be conducted by three researchers who have experience in qualitative research. Transcripts will be analyzed using inductive content analysis [48] with the aid of ATLAS.ti 8.4.26.0 software. A coding sheet will be developed following an iterative process of discussion among the research team. Transcripts will be read carefully several times, and a summary of excerpts will be generated and coded. All transcripts will be reread as new codes develop. Similar codes will be gathered into categories, and similar categories into themes. The themes will be illustrated by the interviewees’ quotations, which will be identified by an alphanumeric code.

### 2.4. Data Management

All data will be stored on a password-protected computer and a central hard disk from the research team. The data of the participants will be pseudonymized, and only the principal investigator will be able to link raw data to personal data. Data will be stored for a minimum of 5 years after the appearance of publications based on this database. After this period, the data can be destroyed.

### 2.5. Public Involvement

A representative of the Piedmont Alzheimer’s Association and a representative of the Light For Life Onlus are part of the Steering Committee of the project and will be part of the interdisciplinary panel responsible for co-designing the booklet. 

## 3. Discussion

Communication skills have been identified among the potentially modifiable factors that could be targeted to reduce burdensome care in the NH setting [39,49]. Indeed, thorough and timely communication about poor prognosis and disease progression contributed to providing palliative-oriented care to NH residents; instead, when communication was absent, delayed, or poor, the provision of curative-oriented care was more likely [5]. However, to our knowledge, so far, no work has been conducted to understand the effect of communication skills training programs targeted at NH staff on the quality of communication and the residents’ intensity of care. The *Teach-to-Communicate* intervention aims to address this gap of knowledge. 

Our training program will be awarded credits for medical education as recommended by experts [24] and is based on an innovative pedagogical approach that combines the learning of communication skills with the promotion of relational abilities and reflection [27]. We believe that educational interventions aimed at improving communication skills should simultaneously focus on modifying behaviors and promoting personal development, since good communication is both a skill and a way of being in relation to the other [50]. 

Consistent with the recommendations of experts in the field of communication skills training [24] and the common features of the most efficacious educational programs in the fields of health and social care [32], our training program will combine classroom-based theory with experiential learning. Theoretical content underpins experiential learning, which in turn is a key driver for behavioral change [32]. Moreover, e-learning self-education activities with the opportunity for facilitated online discussions will be an integral part of the training program. This fulfils the need for more attractive and innovative initiatives to deliver communication skills training beyond traditional workshops [14,24]. We will rely on several didactic methods to sustain effective learning, such as discussion in small groups to allow active participation and didactic videos with trained actors that show good and poor communication practices supported by scripts and critical commentary to stimulate reflection [32]. In addition, written material (i.e., a communication booklet) will be available to provide guidance in care practice. Reflective practice represents a pivotal and constant component of our training program: group discussion will take place after watching videos, experiential learning, and e-learning activities. Indeed, reflection supports the meaning-making of the experience [51] and helps with flexibility in navigating serious illness conversations [52]. Lack of structured debriefing processes is often responsible for reduced efficacy of training programs [32].

The optimal length of a training program to promote changes in the attitudes of HCPs is highly debated, even if there is some evidence for a dose-response relationship [24,32]. A total duration of at least 8 h with individual sessions of 2 h or more seems more likely to promote positive attitude change [32]. Experts recommend a course of at least three days to ensure the transfer of skills to clinical practice [24]. Instead, most training programs targeted at NH staff to enhance end-of-life care were very brief and had short follow-up [53]. Our educational intervention will develop over 6 months with a duration of 20 h to provide HCPs with time to engage with the training, consolidate learning, and change their practice. 

Our educational intervention is targeted at HCPs of any profile since we aim to sensitize the overall staff to the issue of challenging conversations. Moreover, the literature recommends multidisciplinary teamwork to ensure person-centered communication and, finally, high-quality care [54]. Previous authors [20] found that interprofessional learning promoted a deeper understanding of each other’s professional roles, specific communication approaches, and a sense of belonging to a collaborative community. Thus, the interdisciplinary nature of our educational intervention is expected to promote reflection on lived experience, improve attitudes toward interprofessional collaboration [55], and sustain the core message that teamwork is essential to successfully manage serious illness conversations [56]. 

The outcome assessment of the educational intervention will include HCPs’ self-assessed performance, self-efficacy, and confidence in sustaining serious illness conversations, since confidence is related to being able to handle challenging conversations [27]. Moreover, we will explore family carers’ self-reported quality of communication to capture a more relevant feature. The QOC questionnaire employed to assess the quality of communication is a validated measure that allows for investigating satisfaction with specific communication features in addition to providing an overall rating [36]. This tool informs about several nuances of communication, including relational aspects (e.g., “feeling given full attention, listened to, cared as a person”), behaviors (“use understandable words, look in eye”), and communication skills (e.g., “asking about what is important, details of getting sicker, or spiritual and religious beliefs). This two-tier assessment paired with the qualitative interviews will highlight mismatched perceptions, if any, and provide insight on aspects of the educational intervention that work well and what may instead be improved (e.g., paying more attention to some skills in the training program). 

Among the outcomes, we will also look at the completion of any ACP documents. This will inform us about the potential effectiveness of our educational intervention on patient-level outcomes and address one of the main gaps in the communication skills training literature [13]. The communication about future care, treatment preferences, values, and goals among HCPs, the person, and their family carer influences the completion of ACP documents [57,58]. At the same time, exploring HCPs’ and family carers’ perceived quality of communication and the occurrence of ACP documentation will provide insight on the quality of the communication process, beyond a simple binary outcome of ACP documentation occurrence. 

### Limitations

A single-arm design for the pilot study was employed since interventions to improve the quality of communication in NHs are currently at a premium and designing a randomized trial was judged ethically unviable. Our training program targeted only HCPs; interventions targeting multiple stakeholders (i.e., HCPs, residents, and/or family carers) were suggested to improve the quality of serious illness conversations [paper under revision]. However, in the current Italian situation regarding the communication skills needs of NH staff, we deemed it a priority to focus on enhancing the communication competencies of HCPs [7]. 

## 4. Ethics and Dissemination

This study has been approved by the Ethics Committee of the University of Torino, Italy (reference number 0675977/2021). Participants involved in the co-creation process of the booklet, the pilot pre-post study, and the nested qualitative study will receive an information letter with all of the details about the study. Each participant will be asked to sign an informed consent after being informed about the study’s purposes and procedures, the right to leave the study at any time, and the data confidentiality.

The dissemination strategy will be based on the Model for Knowledge Transfer and Dissemination (EMTRek) that has already been implemented in palliative care settings [59]. The strategic lines of the dissemination will include the definition of:

(a) messages regarding end-of-life communication will be tailored to the stakeholders through interactive processes that consider the context;

(b) stakeholders identified in: (i) public administrations, to sensitize about the relevance of the project; (ii) family carers of NH residents, lay citizens, and volunteers, to raise awareness about the project; (iii) HCPs at the local and regional levels, to share findings throughout the project period and equip them with the knowledge developed; (iv) associations of patients or families, to enhance their awareness regarding tools and training programs that may benefit the quality of communication in NH; (v) future generations of HCPs, by sharing the project and its findings in undergraduate and post-graduate programs, to inform students on strategies to ameliorate end-of-life care and improve their communication skills; (vi) research communities, by sharing findings through peer-reviewed publications, conferences, and workshops; and (vii) policymakers, to inform them about the policy issues addressed by the project and invite them to consider the recommendations that emerged.

(c) processes and (d) contexts: during the project, meetings at the local and regional levels will be scheduled to disseminate the project and its intermediate findings with public administrators, family carers of NH residents, HCPs of the involved NHs, professional boards, volunteers, lay citizens, and associations of patients or families. Elective courses on strategies to promote high-quality communication will be offered to bachelor’s and master’s degree students. Findings emerged in the pre and post phases will be presented at national and international congresses, and scientific publications will be ensured. A website (associated with other social media platforms, such as Instagram 264.0.0.22.106, Facebook 394.0.0.40.107, and Twitter 9.68.1-release.0), a logo, and a promotional video (Appendix A) will be created to promote networking and share activities, findings, tools, and other outputs of the project.

## 5. Conclusions

This protocol describes the educational pedagogy, structure, implementation, and expected key outcome measures of the *Teach-to-Communicate* intervention (booklet and HCPs training program), an educational program aimed at improving the communication skills of NH staff. 

The *Teach-to-Communicate* intervention will combine the learning of communication theories and skills with the promotion of relational abilities, since serious illness conversations encompass cognitive, behavioral, and affective domains and require relational qualities of empathy, flexibility, self-reflection, and creativity in addition to mastering communication protocols and techniques. This intervention will employ an innovative blended approach that includes classroom-based theory, experiential learning, and e-learning, and rely on interdisciplinary contexts and several didactic methods to stimulate reflection, engender effective learning, and hopefully drive behavioral change. The *Teach-to-Communicate* intervention has the potential to improve person-centered care as it is expected to enhance the quality of communication in NH and HCPs’ confidence in sustaining serious illness conversations, reduce family carers’ psycho-emotional burden and improve their satisfaction with the care received, and increase ACP documentation. 

For future research, we recommend testing the effectiveness of this training program on larger numbers and different settings with some adjustments. 

## Figures and Tables

**Figure 1 ijerph-20-00725-f001:**
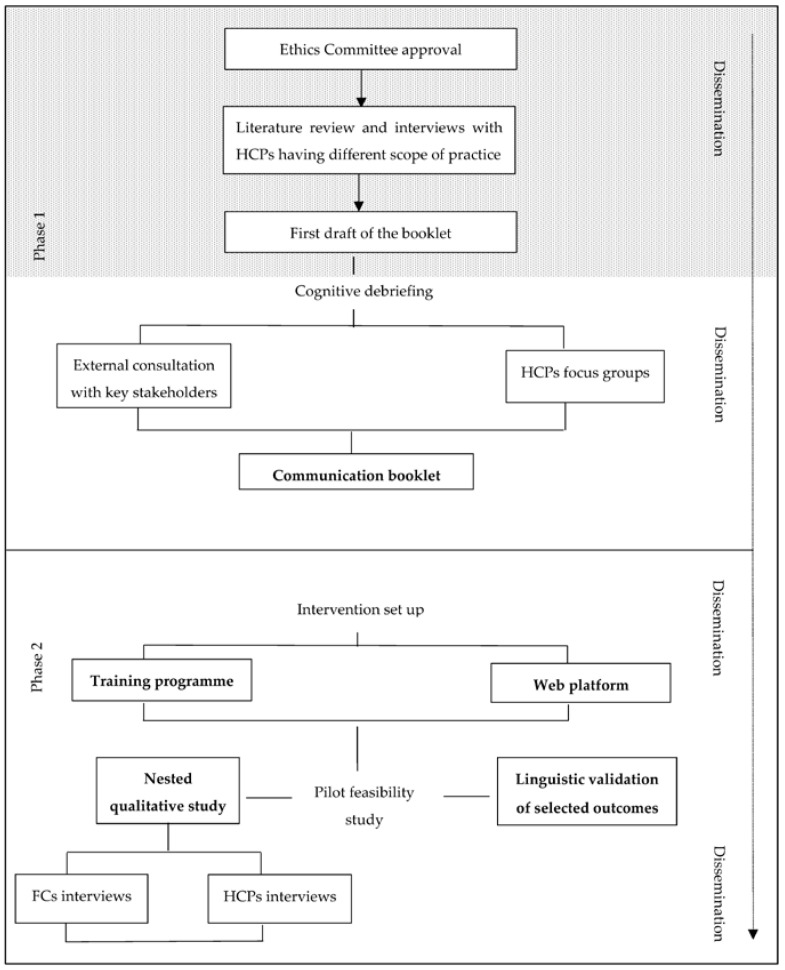
Flow chart of the *Teach-to-Communicate* project. Note. The dot area identifies the advancement status at the time of manuscript submission. Abbreviations. FCs, Family carers; HCPs, Healthcare professionals.

**Figure 2 ijerph-20-00725-f002:**
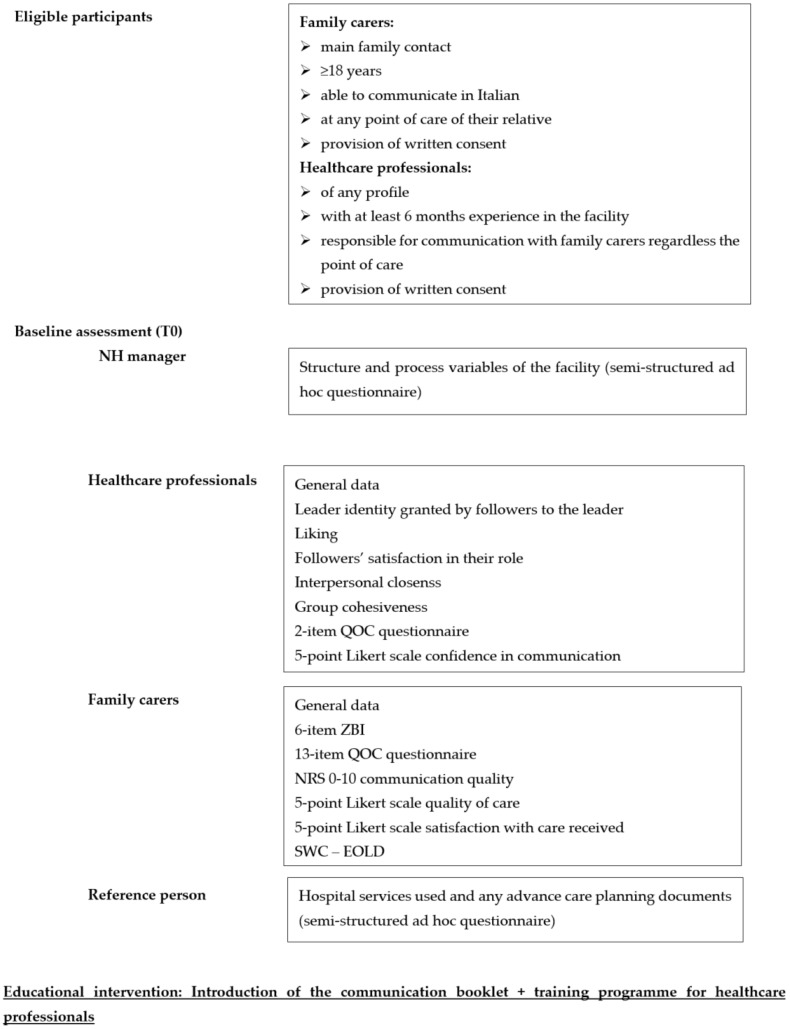
Summary of the study procedure. NRS, numeric rating scale; QOC, quality of communication; SWC-EOLD, satisfaction with care at the end-of-life in Dementia; ZBI, Zarit Burden Interview.

**Table 1 ijerph-20-00725-t001:** Variables aimed at describing the context of the facility (in alphabetical order).

Tool(s)	Assessor	Construct	Author
Followers’ satisfaction in their role	HCPs	Relationship with the leadership	Gatti, Hall & Schyns, 2014 [44]
Interpersonal closeness	HCPs	Relationship with colleagues	Aron, 1992 [41]
Leader Identity granted by followers to the leader	HCPs	Relationship with the leadership	Lord and Gatti [personal communication 2018]
Liking—The Leader Affect Questionnaires	HCPs	Relationship with the leadership	Martinko, Mackey, Moss, Harvey, McAllister, Brees, 2018 [43]
Group cohesiveness	HCPs	Relationship with colleagues	Dobbins & Zaccaro, 1986 [42]
Semi-structured ad hoc questionnaire	NH manager	Structure and process variables of the NH	-

HCPs, healthcare professionals; NH, nursing home.

**Table 2 ijerph-20-00725-t002:** Outcome measures (in alphabetical order).

Tool(s)	Assessor	Construct	Author	Italian Version	Timing
2-item QOC questionnaire	HCPs	Quality of communication	Engelberg et al., 2006 [36]	Solari et al. [in validation]	T0/T1/T2
5-point Likert scale	Family carers	Quality of care	-	-	T0/T1/T2
5-point Likert scale	Family carers	Satisfaction with the care received	-	-	T0/T1/T2
5-point Likert scale	HCPs	Confidence in communication	-	-	T0/T1/T2
NRS 0–10	Family carers	Quality of communication	-	-	T0/T1/T2
13-item QOC questionnaire	Family carers	Quality of communication	Engelberg et al., 2006 [36]	-	T0/T1/T2 *
Semi-structured ad hoc questionnaire	Reference person	Hospital services used, advance care planning documents	-	-	T0/T1/T2
Semi-structured interview	Family carers	Intervention quality, implementation barriers, and suggestions for improvement	-	-	T3
Semi-structured interview	HCPs	Intervention quality, implementation barriers, and suggestions for improvement	-	-	T3
SWC-EOLD	Family carers	Satisfaction with the care received	Volicer et al., 2001 [37]	-	T0/T1/T2
ZBI	Family carers	Family carers burden	Hebert et al., 2000 [45]	Chattat et al., 2011 [46]	T0/T1/T2

HCPs, healthcare professionals; NH, nursing home; QOC, quality of communication; SWC-EOLD, satisfaction with care at the end of life in dementia; ZBI, Zarit Burden Inventory. * The quality of communication perceived by family carers at T2 is the primary outcome of the study.

## Data Availability

The data presented in this study will be available on request from the corresponding authors.

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
