# Peer review of "The Impact of Health and Social Care Professionals’ Education on the Quality of Serious Illness Conversations in Nursing Homes: Study Protocol of a Quality Improvement Project"

_ijerph, 2022, doi:10.3390/ijerph20010725_

Round 1

Reviewer 1 Report

This is an innovative and interesting idea. Nevertheless, the overall study design is chaotic and very difficult to follow. The intervention evaluation also could be better.

From Figure 1 I do not understand why a second ethics committee approval is needed. If the first ethics committee did not approve the whole study then I should not be reviewing; this must be sent to an ethics committee first. I don’t know what am I to suppose or do since several documents like the consent forms are not provided.

I am also unaware if this study was previously reviewed as part of the ethics approval or funding process. No funding is reported by the authors, which is very concerning given the duration and complexity of the study. Do authors have enough resources to go through the whole protocol?

Regarding the training program, there is no defined number of participants and the fact that the course takes 6 months to complete may lead to serious losses. There is also a lot of volunteer bias lurking in the selection process which will limit the internal and external validity of the results.

Outcomes are very much self-reported and this is not a good combination with self-selection and the absence of a control group. Maybe surveying the residents of nursing home could provide some credibility to the effects of the intervention, but that means going back to the drawing board.

Regarding the quantitative analysis, whatever regression is performed requires hierarchical modeling. Moreover, deciding a missing value treatment a priori is like deciding what to eat before you know to which restaurant you are going. In any case, mean imputation is the worst idea since it distorts the marginal distribution and affects the covariance with other variables.

Lastly, I believe authors are missing a huge opportunity by not conducting a randomized trial. I really do not understand the ethical part. If authors think that the current practices  in nursing homes are ethical then I do not see why including a control group is a problem. If they believe they are unethical then again I don’t see the problem since this is the standard treatment and would happen either way, regardless of the intervention.

Some minor comments…

Isn’t there a keyword limit?

line 54. “one month FROM death” I think the authors meant to say.

line 78. Which bias exactly?

Figure 1. typo. nEsted

lines 136–7. Is there evidence for each scenario? 

lines 143–5. What is the length of the booklet? Are 7 days enough? I count 17 participants in the above sentence, why only 5-6 are included in the focus groups? 

Who will asses the booklet? Both family carers and health professionals?

Lines 174–7. How exactly authors plan to validate the questionnaire? The scores on usefulness and feasibility depend heavily on who completes the questionnaire, but I m not  sure who does.

Line 248. Is there a better way than self-certification?

Lines 391—392. Why is the relationship information relevant to this research?

Lines 536—537. Define SD and IQR.

Reviewer 2 Report

The impact of nursing home staff education on the quality of serious illness conversations: study protocol of a quality improvement project

O would like to congratulate the authors for this relevant topis, but maily because it adresses na extraordinary and relavant need on helathcare professionals training.

My congratulations.

the project is very relevant and pertinent. The description of the development of the program is very interesting, clear and coherent. Study designs, instruments used, data analysis are consistent and valid. Ethical considerations are well stated and also data protection.

Abstract:

At the first, the abstract seems to refer to the care of people with frailty. However, in the introduction we realized that, the communication program is centered on end of life conversations, regardless of the condition of NH residents. These issue should be clarified by the authors.

The target population of this programm is not well understood, namely at the title. Are these staff only healthcare profissionals or others? If so, i believe that it could be well stated at the title.

Introduction: The theme is aparently well stated, and the theoretical framewrok is clearly presenteed. The need for the development of this programm is also well adressed and what the innovation that it and the innovation that adds.

Aims: Are the authors only interested in the quality of communication reported by the family? and what about the residentes experience with communication ability from HCP’s?  In this sense, the introduction should clarify it.

Research methods and analysis

The authors refered in study design, some guidelines that will be used to report information. However, information about the design of the study should be stated. Authors referes to Standard Protocol Items. A reference should be inserted.

p.271 –  in the line “The main steps will be as follows:” A reference should be inserted.

Line 447. Quality of communication - Why does the opinion of residents will not be considered?
